# Mild and moderate COVID-19 during Alpha, Delta and Omicron pandemic waves in urban Maputo, Mozambique, December 2020-March 2022: A population-based surveillance study

Brecht Ingelbeen[1,2☯]*, Victória Cumbane[3☯], Ferão Mandlate[3], Barbara Barbé[1], Sheila Mercedes Nhachungue[3], Nilzio Cavele[3], Cremildo Manhica[3], Catildo Cubai[3], Neusa Maimuna Carlos Nguenha[3], Audrey Lacroix[4], Joachim Mariën[1], Anja de Weggheleire[5], Esther van Kleef[1,2], Philippe Selhorst[1], Marianne A. B. van der Sande[1,2], Martine Peeters[4], Marc-Alain Widdowson[1], Nalia Ismael[3‡], Ivalda Macicame[3‡]

1 Instituut voor Tropische Geneeskunde, Antwerp, Belgium, 2 Julius Center for Health Sciences and Primary Care, Utrecht University, Utrecht, The Netherlands, 3 Instituto Nacional de Saúde, Ministry of Health, Marracuene, Mozambique, 4 TransVIHMI (Recherches Translationnelles sur VIH et Maladies Infectieuses), Université de Montpellier, Institut de Recherche pour le Développement, INSERM, Montpellier, France, 5 Médecins sans Frontières, Brussels, Belgium

☯ These authors contributed equally to this work.
‡ These authors also contributed equally to this work.
* brechtingelbeen@gmail.com

**Data Availability Statement:** Pseudonymized data supporting the findings of this study/publication

## Abstract

In sub-Saharan Africa, reported COVID-19 numbers have been lower than anticipated, even when considering populations' younger age. The extent to which risk factors, established in industrialised countries, impact the risk of infection and of disease in populations in sub-Saharan Africa, remains unclear. We estimated the incidence of mild and moderate COVID-19 in urban Mozambique and analysed factors associated with infection and disease in a population-based surveillance study. During December 2020-March 2022, 1,561 households (6,049 participants, median 21 years, 54.8% female, 7.3% disclosed HIV positive) of Polana Caniço, Maputo, Mozambique, were visited biweekly to report respiratory symptoms, anosmia, or ageusia, and self-administer a nasal swab for SARS-CoV-2 testing. Every three months, dried blood spots of a subset of participants (1,412) were collected for detection of antibodies against SARS-CoV-2 spike glycoprotein and nucleocapsid protein. Per 1000 person-years, 364.5 (95%CI 352.8–376.1) respiratory illness episodes were reported, of which 72.2 (95%CI 60.6–83.9) were COVID-19. SARS-CoV-2 seroprevalence rose from 4.8% (95%CI 1.1–8.6%) in December 2020 to 34.7% (95%CI 20.2–49.3%) in June 2021, when 3.0% were vaccinated. Increasing age, chronic lung disease, hypertension, and overweight increased risk of COVID-19. Older age increased the risk of SARS-CoV-2 seroconversion. We observed no association between socio-economic status, behaviour and COVID-19 or SARS-CoV-2 seroconversion. Active surveillance in an urban population confirmed frequent COVID-19 underreporting, yet indicated that the large

are retained at the Institute of Tropical Medicine, Antwerp and can be made available after approval of a motivated and written request to ITMresearchdataaccess@itg.be. Study protocol, scripts for conducting the analysis, and anonymized data, without geo-located or other data that could allow identification, are available on https://github.com/ingelbeen/africover-git.

**Funding:** The work was funded by the European & Developing Countries Clinical Trials Partnership (EDCTP, RIA2020EF-3031 to BI). The funders had no role in study design, data collection and analysis, decision to publish, or preparation of the manuscript.

**Competing interests:** The authors have declared that no competing interests exist.

majority of cases were mild and non-febrile. In contrast to reports from industrialised countries, social deprivation did not increase the risk of infection nor disease.

## Introduction

In sub-Saharan Africa, reported COVID-19 cases and deaths were lower than expected from estimated SARS-CoV-2 seroprevalence in the region, and age-specific infection fatality [1,2]. Older age, deprivation, black ethnicity (compared to white) and co-morbidities have been found to independently increase the risk of COVID-19 disease or death in different contexts [3–5]. Younger age groups less frequently manifested symptoms or were hospitalised when infected, but also had lower infection rates [6,7]. HIV increased the risk of COVID-19-related death [8], as did comorbidities as hypertension, diabetes, or chronic kidney of pulmonary disease [3]. Higher COVID-19 incidence among deprived populations has been hypothesised to result from inequalities in the ability to work remotely, and by higher secondary infection rates within (more crowded) households [9].

The extent to which non-pharmaceutical interventions had an impact on COVID-19 incidence in sub-Saharan African settings is still poorly understood. Other studies in Eastern Africa have shown a reduction in excess mortality among children, indicating an effect of social restrictions on pathogen spread [10].

In Mozambique (estimated pop. 2020 31.2 million), 184,219 COVID-19 cases and 2010 deaths were reported during 2020–21, yet excess deaths due to the pandemic have been estimated at 78,100 (95%CI 54,100–109,000) [11]. The excess mortality rate (139 per 100,000, 95% CI 96–194) was comparable to the global all-age estimate (120 per 100,000, 95%CI 113–129). COVID-19 vaccination started on 7 March 2021. By 8 September 2021, 5.0% of the Mozambican population had received at least one dose of vaccine; by 8 March 2022, 40.2% had.

In Maputo City (estimated pop. 2020 1.1 million), the capital of Mozambique, in 2020–21, COVID-19 testing was centralised in two COVID-19 management facilities, but mild/moderate cases were unlikely to go there. Here, we provide a comprehensive description of the clinical range of mild and moderate COVID-19 in urban Maputo. We then estimate SARS-CoV-2 (infection) seroprevalence, COVID-19 (disease) incidence rates, and analyse demographics, comorbidities, and exposures increasing the risk of SARS-CoV-2 seroconversion and of COVID-19.

## Methods

### Study design and population

Between 15 December 2020 and 31 March 2022, population based surveillance in urban Maputo consisted of two components: biweekly follow-up of households to record possible COVID-19 cases and three-monthly sero-surveys to track SARS-CoV-2 antibodies in a subset of participants. Recruited households were embedded in the Health and Demographic Surveillance System (HDSS) of Polana Caniço, covering 15,393 residents.

### Sampling strategy

Study interviewers were each assigned a housing block for which they received a list of households enrolled in the HDSS. Eligible households were recruited consecutively following informed consent by the household head and by individual household members randomly

selected for the repeated sero-survey. To participate in the study, household members had to be residing in the household since ≥3 months, including all ages. Subsequently, households were visited or phoned every two weeks during one year to detect possible COVID-19 cases at the time of the visit or with symptom onset in the two weeks prior to the visit. Household enrolment stopped after three months. After that, biweekly follow-up visits continued for a year and the second sero-survey round started.

## Data collection

At baseline, household-level water- and sanitation conditions, individual weight and height (measured during a visit to determine Body Mass Index, BMI, in ≥16 year olds), self-reported comorbidities, smoking, and behaviour potentially determining exposure risk (e.g., use of public transportation, healthcare occupation, crowding) were recorded in electronic questionnaires. Within the HDSS, participating household members' demographics, educational level, and households' socio-economic quintile (relative to wealth across HDSS households) were previously recorded and last updated in 2019. During the biweekly household follow-up visits, possible COVID-19 cases in the household were registered, recording clinical signs and symptoms, potential exposures (according to the World Health Organization's Revised case report form for Confirmed Novel Coronavirus COVID-19, accessed on 4 June 2020), and prior COVID-19 vaccination. A possible COVID-19 case was any respiratory sign or loss of smell or taste, anosmia, or ageusia, with or without fever, and symptom onset in the previous two weeks, among any household member. Possible COVID-19 cases were asked to self-administer a nasal swab for SARS-CoV-2 PCR testing, either–if present–during the household visit, or–if absent–during an ad hoc interviewer visit scheduled shortly after. Household contacts of possible cases were not tested. If COVID-19 was confirmed, the case was followed up after 28 and 56 days to record clinical outcome.

For the repeated sero-survey, randomly selected participating household members from three age strata (0–17, 18–49, and ≥50 years) were visited every three months during one year to collect dried blood spots from a finger prick and (from 31 March 2021 onwards) record prior COVID-19 vaccination.

## Laboratory procedures

To confirm COVID-19 in possible cases detected within the biweekly follow up of households, nasal swabs were self-administered at cases' homes or assisted by interviewers in cases under 5 years of age, then transported in RNA Shield™ reagent. Real-time reverse-transcription PCR was performed within the same day at Instituto Nacional de Saúde, Marracuene, Mozambique.

To determine SARS-CoV-2 seropositivity within the repeated sero-surveys, dried blood spots were collected, containing 450μl of blood sampled from a fingerprick in six circles (each approximately 75ul) on dried blood spot filter paper (Whatman 903™ Protein Saver Card). Samples were prepared for testing by punching two discs of 4mm diameter (corresponding to 40 μl of blood), and eluted overnight in 160 μL of hypertonic phosphate buffered saline-BSA (dilution 1:40, phosphate buffered saline-1% BSA-0.15% Tween, pH 7.4, Sigma-Aldrich). Before use in the immunoassay, eluted samples have been further diluted to 1:200 in hypertonic phosphate buffered saline-BSA, according to Mariën et al[12]. We then used an in-house developed multiplex antibody assay for the detection of anti-SARS-CoV-2 IgG. We coupled recombinant large spike glycoprotein S1 and S2 subunit, receptor-binding domain (RBD), and nucleocapsid-protein (NP) antigens derived from SARS-CoV-2 at Sino Biological to maximum of $1.25 \times 10^6$ paramagnetic MAGPLEX COOH-microsphere beads from Luminex

Corporation as antibody targets. 150 μl of beads and diluted sera were added to each well, incubated at room temperature, then washed with 200 μl/well of hypertonic phosphate buffered saline-BSA. Adding biotin-labelled anti-human secondary IgG and streptavidin-R-phycoerythrin conjugate, another 30 min incubation, samples were read on Luminex MagPix$^{TM}$ at Instituto Nacional de Saúde, Marracuene, Mozambique. For each antigen target, a cut-off value of antibody detection was estimated by adding 2.5 standard deviations to the average value of 42 negative control samples from residents of Maputo, collected prior to the pandemic and also spotted on filter paper. We used two criteria to determine seropositivity: (i) both RBD and NP above the cut-off, ensuring excellent specificity as demonstrated in [12], and (ii) RBD above the cut-off, similar to assays used in most other SARS-CoV-2 sero-surveys [7]. After breakdown of the MagPix$^{TM}$ platform at Instituto Nacional de Saúde, serological testing for samples collected after August 2021 became unavailable.

Dried blood spot cards containing unused blood spots and aliquots of nasal swab samples were stored in -80˚C freezers at Instituto Nacional de Saúde and will be destroyed after five years.

## Data analysis

We estimated incidence rates of acute respiratory illness and of COVID-19 from respectively the number of possible COVID-19 cases and of confirmed COVID-19 cases, divided by the observation time. Because every household visit recorded possible cases with onset during two weeks before the visit, observation time consisted of the two weeks prior to each visit, or the time between visits if less than two weeks spanned between consecutive visits. We analysed clinical signs and symptoms associated with COVID-19, comparing confirmed cases to possible cases testing negative for SARS-CoV-2, adjusting for age using unconditional logistic regression.

To identify participant demographic, health, socio-economic and behavioural characteristics associated with COVID-19, we did a survival analysis fitting a Cox proportional hazards model with self-reported first confirmed COVID-19 as event variable, and observation time, censored after a first confirmed COVID-19 episode, as time variable, adjusting for age and sex.

We estimated infection- and vaccination-induced SARS-CoV-2 seroprevalence, by age group, based on sero-survey samples collected up to 31 July 2021. Samples were SARS-CoV-2 sero-positive when antibodies against RBD and NP were detected, as proposed by a validation study of the assay [12]. To ensure comparability to results with other sero-surveys, we also analysed seroprevalence based on antibodies against RBD only.

To identify participant characteristics associated with SARS-CoV-2 infection (including asymptomatic), similar to the above survival analysis of first symptomatic COVID-19, we fitted a Cox proportional hazards model to sero-survey participants with ≥ 2 samples collected up to 31 July 2021. SARS-CoV-2 infection (event variable) was defined as either a SARS-CoV-2 positive result following a negative result (sero-conversion), or an initial SARS-CoV-2 positive result. The time variable consisted of three months prior to each sero-survey, or the time between consecutive sero-surveys if less than three months spanned in-between, and was–in case of sero-conversion–censored at the midpoint between the last negative test and the subsequent positive test.

## Ethical considerations

The study protocol and amended protocol (adding COVID-19 vaccination to case report forms) were approved by Institutional Review Boards of the Institute of Tropical Medicine and of Instituto Nacional de Saúde, the National Committee for Bioethics in Health of

Mozambique (CNBS; Comité Nacional De Bioética para Saúde, 517/CNBS/2020) and the Antwerp University Hospital ethics committee (B3002020000123). The study obtained administrative approval by the Minister of Health and the Health authorities of the Municipal Council, through cover letters of the study. Study participants provided written informed consent at baseline for study participation, and again at the time of collecting a nasal swab or at the first sero-survey visit. For minors, written informed consent was asked to parents or guardians, on top of a written assent form for 12 to 18 year old participants. In case of illiteracy, a fingerprint from the participant and a signature from a witness (person ≥18 years old, not part of the study team) were obtained.

## Results

### Household surveillance of acute respiratory illness

Between 15 December 2020 and 31 March 2021, we conducted 11,925 household visits in 1,561 households, covering 6,049 participants (Fig 1). Participants were median 21 years old (interquartile range, IQR, 11–38 years), 3,315 (54.8%) were female, and 435 (7.3%) disclosed to be HIV positive. 2,694 (55.6% of 4,841 with recorded socio-economic status) did not complete primary education and 93 (1.9%) had higher education.

### Respiratory illness and COVID-19 incidence rate

In the two weeks prior to the visits, 482 (30.9%) households reported at least one possible case in the household. The incidence rate of respiratory illness was 364.5 (95% CI 352.8–376.1) per

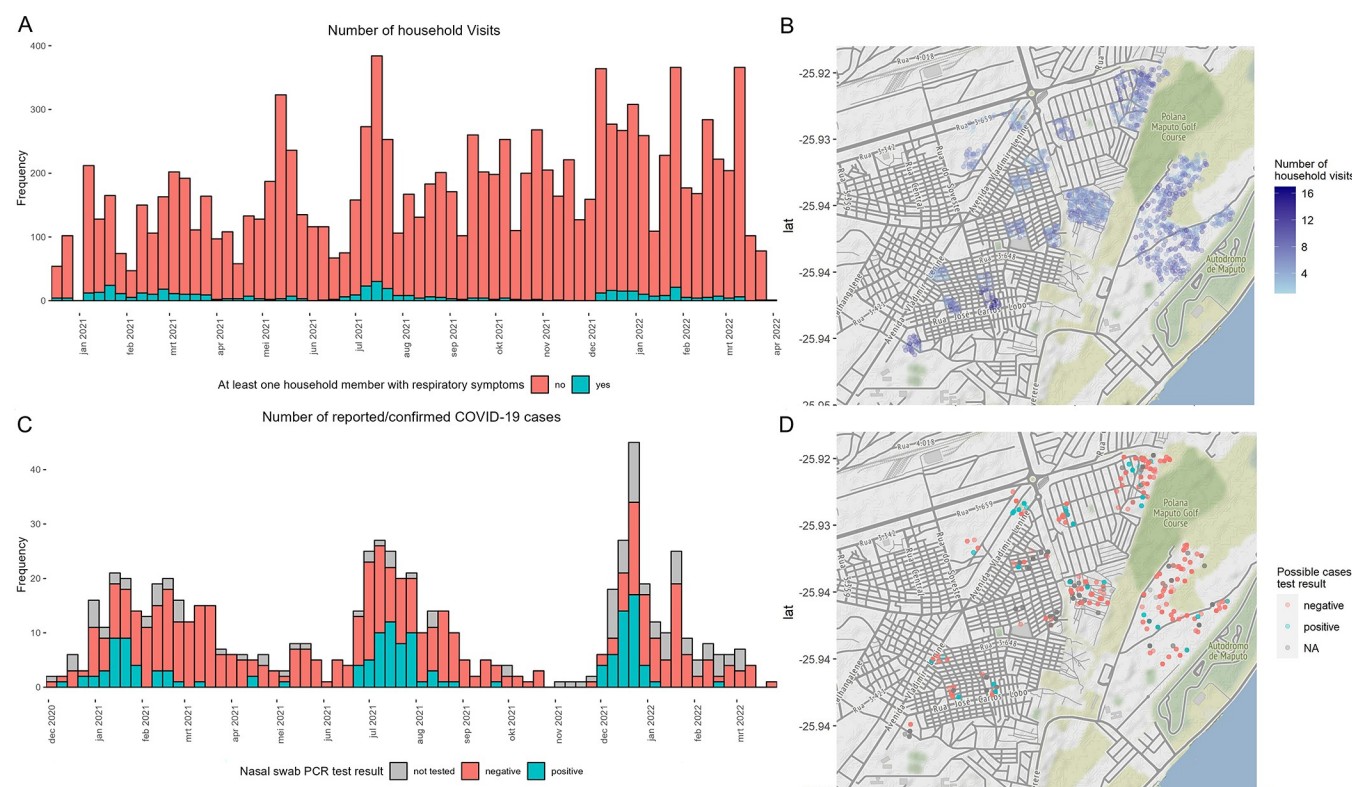

**Fig 1. Household acute repiratory illness and COVID-19 surveillance.** A. Weekly frequency of household visits, B. Geographical distribution of household visits in Maputo City, C. Weekly number of possible COVID-19 cases, stacked by SARS-CoV-2 PCR test result, D. Geographical distribution of possible COVID-19 cases by result. Possible cases were not tested if a nasal swab could not be correctly collected and transported to the reference laboratory. Maps were made in R using the ggmap package and map tiles by Stamen Design, under CC BY 4.0. Data by OpenStreetMap, under ODbL.

1,000 person-years (py, 691 possible cases in 611 participants; 1,895.9 py followed up). Of 579 possible cases, a nasal swab was collected and tested, median 5 days (IQR, 3–8 days) after symptom onset. SARS-CoV-2 was confirmed in 144 (24.9%) cases. The incidence rate of confirmed COVID-19 was 72.2 (95% CI 60.6–83.9) per 1,000 py. Among participants under 18 years old, this was 25.9 (95%CI 15.0–36.8) per 1,000 py; in 18–49 year olds it was 79.9 (95%CI 61.2–98.5) per 1,000 py; in ≥50 year olds, it was 188.3 (95%CI 141.8–234.9) per 1,000 py. SARS-CoV-2 positivity of tested possible cases peaked at 38.2% in January 2021, at 42.2% in July 2021, and at 53.1% in December 2021.

## Clinical signs and symptoms of mild and moderate COVID-19

Reported COVID-19 cases were median 36.4 years old (IQR 22.3–57.5 years) and 87 (60.4%) were female (Table 1). Compared to SARS-CoV-2 negative cases (median age 26.0 years, IQR 10.8–49.3, 55.8% female), COVID-19 cases had more frequently anosmia (age-adjusted odds ratio, aOR 2.36 95%CI 1.48–3.58), ageusia (aOR, 2.29 95%CI 1.45–3.58), loss of appetite (aOR 2.21 95%CI 1.37–3.56), and chills (aOR 1.78 95%CI 1.05–2.97). During the Omicron variant wave starting December 2021, the association with each of these symptoms disappeared (anosmia aOR 1.1 95%CI 0.46–2.56, ageusia 0.70 95%CI 2.29–1.62, loss of appetite 1.25 95%CI 0.52–2.96, chills 0.88 95%CI 0.28–2.49).

## Characteristics associated with symptomatic COVID-19

Increasing age (in ≥70 year olds hazard ratio (HR) 10.2, 95%CI 3.58–29.09) and several reported comorbidities increased the risk of symptomatic COVID-19: leukaemia, chronic lung disease, overweight, and hypertension (Table 2). We found no increased risk of COVID-19 in people with HIV (HR 0.96, 95%CI 0.45–2.04) or with (a history of) tuberculosis (HR 1.06, 95% CI 0.38–2.96). Households with no or limited handwashing facilities members (HR 0.54, 95% CI 0.34–0.87) and with bedrooms shared between 3 or more household members (HR 0.45, 95%CI 0.26–0.76), were less likely to report COVID-19.

## Infection-induced SARS-CoV-2 seroprevalence

2185 samples collected until 31 July 2021 of 1412 sero-survey participants (median age 30.6 years, IQR 13.7–57.6; 38.2% ≥50 years old; 55.2% female) were tested. 301 participants (21.3%) tested positive (antibodies against RBD and NP) at least once. 34 (45.3%) out of 75 with a test subsequent to positive test, sero-reverted.

Crude seroprevalence increased from 4.8% (95%CI 1.1–8.6) in December 2020 to 34.2% (95%CI 23.4–45.1) in June 2021, when 2.7% of participants reported vaccination with at least one dose of vaccine (Fig 2). Seroprevalence increased strongest in ≥50 year olds, peaking at 51.6% (95%CI 34.0–69.2) in June 2021, when 3.2% was vaccinated, and then declined to 41.6% (95%CI 26.5–56.7) in July 2021, when 10.9% was vaccinated.

Crude seroprevalence based on antibodies against RBD only was higher, rising from 10.5% (95%CI 5.1–15.9%) in December 2020 to 46.5% (95%CI 36.7–56.3) in July 2021, without decrease in seroprevalence from June to July 2021 (S1 Table).

## Characteristics associated with SARS-CoV-2 infection

Older age increased the risk of SARS-CoV-2 infection (HR 60–69 versus 0–9 years 1.57, 95% CI 1.03–2.39, Table 3). We found no association between SARS-CoV-2 infection and socio-economic, behavioural factors, nor comorbidities.

**Table 1. Clinical signs and symptoms associated with SARS-CoV-2 confirmation among acute respiratory illness (possible COVID-19 cases) reported during December 2020-March 2022, using unconditional logistic regression to estimate age-adjusted and crude odds ratios.**

| Factors | Confirmed SARS-CoV-2 (N = 144) | | Negative SARS-CoV-2 (N = 435) | | Crude odds ratio | | Age-adjusted odds ratio | |
|---|---|---|---|---|---|---|---|---|
| | n | % | n | % | OR | 95%CI | aOR | 95% CI |
| Age*: 0–9 years | 7 | 4.9 | 90 | 21.5 | ref | | | |
| 10–19 years | 18 | 12.5 | 79 | 18.9 | **2.93** | **1.21–7.87** | | |
| 20–29 years | 34 | 23.6 | 62 | 14.8 | **7.05** | **3.10–18.3** | | |
| 30–39 years | 18 | 12.5 | 48 | 11.5 | **4.82** | **1.95–13.2** | | |
| 40–49 years | 14 | 9.7 | 37 | 8.8 | **4.86** | **1.87–13.8** | | |
| 50–59 years | 24 | 16.7 | 46 | 11.0 | **6.71** | **2.82–17.9** | | |
| 60–69 years | 22 | 15.3 | 40 | 9.5 | **7.07** | **2.92–19.1** | | |
| 70+ years | 7 | 4.9 | 17 | 4.1 | **5.29** | **1.62–17.4** | | |
| Sex*: Female | 87 | 60.4 | 234 | 55.8 | 1.21 | 0.82–1.78 | 1.19 | 0.81–1.77 |
| Clinical signs/symptoms | | | | | | | | |
| Cough | 117 | 89.3 | 326 | 86.0 | 1.36 | 0.75–2.63 | 1.52 | 0.82–2.95 |
| Headache | 81 | 61.8 | 217 | 57.3 | 1.21 | 0.81–1.82 | 1.14 | 0.75–1.73 |
| Rhinorrhoea | 69 | 52.7 | 221 | 58.3 | 0.80 | 0.53–1.19 | 0.86 | 0.57–1.29 |
| Sore throat | 54 | 41.2 | 131 | 34.6 | 1.33 | 0.88–1.99 | 1.22 | 0.80–1.84 |
| Ageusia | 47 | 35.9 | 69 | 18.2 | **2.51** | **1.61–3.91** | **2.29** | **1.45–3.58** |
| Oxygen saturation <95% | 16/48 | 33.3 | 36/151 | 23.8 | 1.60 | 0.78–3.22 | 1.54 | 0.74–3.14 |
| Anosmia | 43 | 32.8 | 63 | 16.6 | **2.45** | **1.55–3.85** | **2.36** | **1.48–3.75** |
| Fatigue | 41 | 31.3 | 95 | 25.1 | 1.36 | 0.88–2.10 | 1.26 | 0.80–1.96 |
| Loss of appetite | 39 | 29.8 | 62 | 16.4 | **2.17** | **1.36–3.44** | **2.21** | **1.37–3.56** |
| Fever | 33 | 25.2 | 96 | 25.3 | 0.99 | 0.62–1.56 | 1.10 | 0.68–1.75 |
| Chills | 29 | 22.1 | 50 | 13.2 | **1.87** | **1.12–3.10** | **1.78** | **1.05–2.97** |
| Joint pain | 29 | 22.1 | 49 | 12.9 | **1.91** | **1.14–3.17** | 1.59 | 0.90–2.59 |
| Chest pain | 25 | 19.1 | 60 | 15.8 | 1.25 | 0.74–2.08 | 1.10 | 0.64–1.85 |
| Myalgia | 24 | 18.3 | 49 | 12.9 | 1.51 | 0.87–2.56 | 1.24 | 0.70–2.13 |
| Nausea | 10 | 7.6 | 23 | 6.1 | 1.28 | 0.57–2.69 | 1.29 | 0.57–2.77 |
| Dyspnoe | 9 | 6.9 | 36 | 9.5 | 0.70 | 0.31–1.44 | 0.62 | 0.27–1.31 |
| Diarrhoea | 8 | 6.1 | 26 | 6.9 | 0.88 | 0.37–1.92 | 0.86 | 0.35–1.89 |
| Vomit | 7 | 5.3 | 19 | 5.0 | 1.07 | 0.41–2.50 | 1.32 | 0.49–3.23 |
| Rash | 3 | 2.3 | 3 | 0.8 | 2.94 | 0.54–16.0 | 2.98 | 0.54–16.4 |
| Nose bleeding | 1 | 0.8 | 3 | 0.8 | 0.95 | 0.05–7.50 | 0.99 | 0.05–7.94 |
| Change of consciousness | 0 | 0.0 | 3 | 0.8 | | | | |

* age and sex missing of 16 possible cases (all SARS-CoV-2 negative), clinical signs and symptoms missing of 69 possible cases (56 SARS-CoV-2 negative, 13 positive). Of 92 confirmed COVID-19 cases followed up after 28 days, of whom 54 again after 56 days, one (1.1%) died.

## Discussion

Population-based COVID-19 surveillance in an urban population cohort in Mozambique confirmed that even during the acute phase of the pandemic the large majority of SARS-CoV-2 infections were asymptomatic, symptomatic cases were mild, and three in four were non-febrile.

Three COVID-19 peaks were distinct and died out after a few weeks, while other respiratory illness (SARS-CoV-2 negative cases) continued to be reported throughout 2021. Sudden drops in COVID-19 incidence after peaks while respiratory pathogens other than SARS-CoV-2 continue to circulate, indicate that newly introduced COVID-19 variants (Alpha, Delta,

**Table 2. Demographic, socio-economic and behavioural characteristics associated with first confirmed COVID-19 among among acute respiratory illness (possible COVID-19 cases) reported in population-based surveillance during December 2020-March 2022, using a Cox regression model adjusting for age and sex.**

| Characteristic | | Cohort participants | | Reported first COVID-19 | | COVID-19 age-/sex-adjusted hazard ratio | |
|---|---|---|---|---|---|---|---|
| | | N | % | N | % | HR | 95%CI |
| Age | 0–9 years | 1331 | 22.0 | 7 | 5.1 | ref | |
| | 10–19 | 1570 | 26.0 | 18 | 13.1 | 2.17 | 0.91–5.20 |
| | 20–29 | 1128 | 18.6 | 30 | 21.9 | **5.01** | **2.20–11.40** |
| | 30–39 | 644 | 10.6 | 17 | 12.4 | **5.13** | **2.13–12.37** |
| | 40–49 | 519 | 8.6 | 14 | 10.2 | **5.03** | **2.03–12.47** |
| | 50–59 | 406 | 6.7 | 24 | 17.5 | **11.05** | **4.76–25.65** |
| | 60–69 | 318 | 5.3 | 20 | 14.6 | **11.81** | **4.99–27.93** |
| | ≥70 | 133 | 2.2 | 7 | 5.1 | **10.20** | **3.58–29.09** |
| Sex | Male | 2804 | 46.4 | 55 | 40.1 | ref | |
| | Female | 3245 | 53.6 | 82 | 59.9 | 1.18 | 0.84–1.67 |
| Socio-economic quintile*$ | 1$^{st}$ (lowest) | 429 | 14.6 | 5 | 4.8 | **0.22** | **0.09–0.57** |
| | 2$^{nd}$ | 579 | 19.7 | 24 | 22.9 | 0.83 | 0.50–1.39 |
| | 3$^{rd}$ | 591 | 20.1 | 12 | 11.4 | 0.38 | 0.20–0.72 |
| | 4$^{th}$ | 640 | 21.7 | 26 | 24.8 | 0.80 | 0.48–1.31 |
| | 5$^{th}$ (highest) | 706 | 24.0 | 38 | 36.2 | ref | |
| Education*$ | None | 1350 | 41.3 | 53 | 47.3 | 0.96 | 0.44–2.13 |
| | Primary | 1557 | 47.6 | 48 | 42.9 | 1.06 | 0.50–2.26 |
| | Secondary | 276 | 8.4 | 8 | 7.1 | ref | |
| | Higher | 88 | 2.7 | 3 | 2.7 | 1.04 | 0.28–3.95 |
| Health worker$ | | 130 | 3.8 | 4 | 5.9 | 1.46 | 0.53–4.02 |
| Reported comorbidities | HIV | 450 | 7.5 | 8 | 10.0 | 0.96 | 0.45–2.04 |
| | (history of) tuberculosis | 179 | 3.0 | 4 | 5.1 | 1.06 | 0.38–2.96 |
| | hypertension | 583 | 9.7 | 22 | 27.8 | **1.91** | **1.05–3.48** |
| | diabetes | 68 | 1.1 | 3 | 3.8 | 1.78 | 0.55–5.82 |
| | asthma | 291 | 4.9 | 4 | 5.1 | 1.16 | 0.42–3.19 |
| | chronic lung disease | 37 | 0.6 | 4 | 5.1 | **8.11** | **2.91–22.58** |
| | chronic heart disease | 54 | 0.9 | 2 | 2.5 | 1.90 | 0.46–7.85 |
| | leukaemia | 4 | 0.1 | 1 | 1.3 | **33.47** | **4.48–249.94** |
| BMI# | underweight | 795 | 37.2 | 12 | 15.8 | 1.45 | 0.68–3.13 |
| | normal | 830 | 38.8 | 27 | 35.5 | ref | |
| | overweight | 281 | 13.1 | 19 | 25.0 | **1.87** | **1.02–3.44** |
| | obesity | 231 | 10.8 | 18 | 23.7 | **2.09** | **1.08–4.06** |
| Smoking | non smoker | 5738 | 95.8 | 74 | 93.7 | ref | |
| | (ex-)smoker | 251 | 4.2 | 5 | 6.3 | 1.10 | 0.43–2.85 |
| Public transportation | none | 3948 | 66.0 | 39 | 49.4 | ref | |
| | bus/train | 2013 | 33.7 | 40 | 50.6 | 1.24 | 0.78–1.96 |
| | moto taxi/shared taxi | 19 | 0.3 | 0 | 0.0 | | |
| Sharing bedroom | 1–2 pers. | 3311 | 55.6 | 61 | 77.2 | ref | |
| | 3 or more pers. | 2646 | 44.4 | 18 | 22.8 | **0.45** | **0.26–0.76** |
| Toilet | in the house | 5061 | 85.0 | 64 | 81.0 | ref | |
| | shared between households | 896 | 15.0 | 15 | 19.0 | 1.42 | 0.81–2.49 |
| Handwash facility | sink/faucet | 1748 | 29.3 | 35 | 44.3 | ref | |
| | bucket/jar/kettle | 1204 | 20.2 | 9 | 11.4 | **0.39** | **0.19–0.81** |
| | none | 3005 | 50.4 | 35 | 44.3 | **0.54** | **0.34–0.87** |

(*Continued*)

**Table 2.** (Continued)

| Characteristic | | Cohort participants | | Reported first COVID-19 | | COVID-19 age-/sex-adjusted hazard ratio | |
|---|---|---|---|---|---|---|---|
| | | N | % | N | % | HR | 95%CI |
| Water available in house | yes | 2153 | 36.1 | 35 | 44.3 | ref | |
| | no | 3804 | 63.9 | 44 | 55.7 | 0.74 | 0.48–1.16 |

*level of education (of the mother among children and adolescents) and socio-economic level have been measured as part of the HDSS round in 2019 and was available for 4746 household members.

$^§$ in $\geq$ 18 year olds.

$^#$ in $\geq$16 year olds

Omicron) could have quickly spread despite non-pharmaceutical interventions. Nonetheless, transmission slowed when infection-induced seroprevalence (e.g., 11.2% in March 2021) was still lower than the herd immunity threshold anticipated from the Alpha variants' reported transmissibility [13].

Surveillance started in December 2020 when the Beta variant had spread for 3 months through neighbouring South Africa, and two weeks before a surge in cases was detected in facility-based disease surveillance in Mozambique. Infection-induced seroprevalence of 4.8% in December 2020 was lower than the corrected pooled 16.2% reported in other African countries that month [7]. Limited spread of SARS-CoV-2 before January 2021 was also supported by facility-based COVID-19 surveillance and by bed occupancy in COVID-19 management facilities. Even with limited confirmatory testing of cases, neither PCR testing nor hospital bed capacity have been overwhelmed until January 2021, supporting the low seroprevalence observed in December 2020. Limited spread of wild type virus and the Beta variant during the first year of the pandemic–contrasting to neighbouring South Africa and Eswatini–could result from limited seeding through imported cases, limited mobility within the city and within the country, and longer maintained non-pharmaceutical interventions (e.g., workplace closure)–even if less stringent–compared to its neighbours [14]. The eventual surge in cases in January

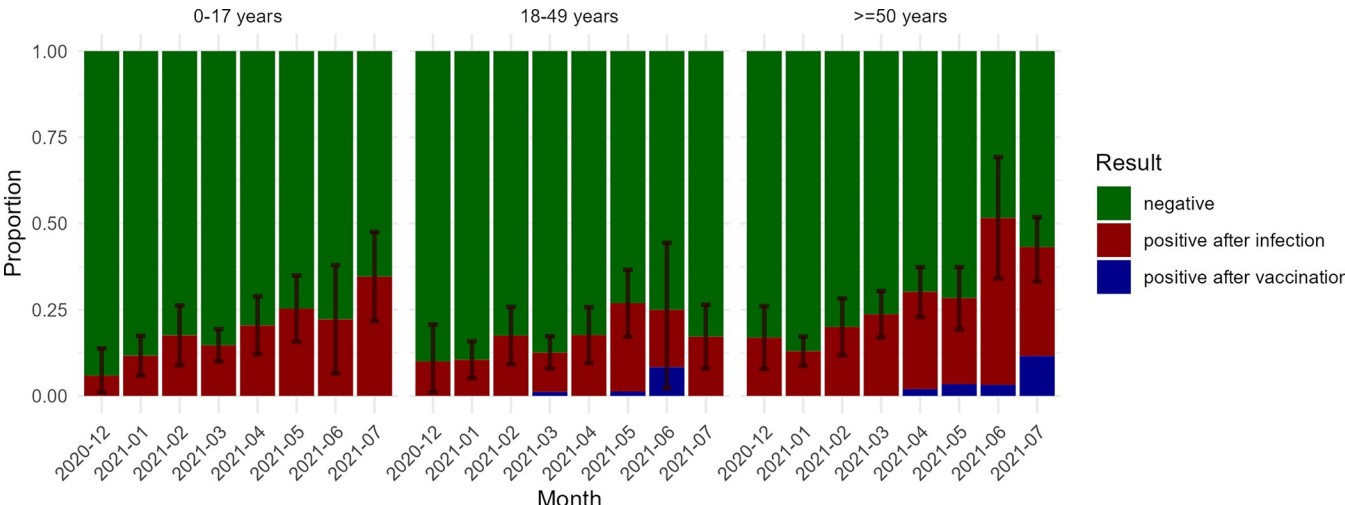

**Fig 2. Infection- and vaccine-induced SARS-CoV-2 seroprevalence by age group, December 2020—July 2021.** $N_{0-17\ years}$ = 647, $N_{18-49\ years}$ = 612, $N_{\geq50\ years}$ = 882. Whether SARS-CoV-2 sero-positivity was infection- or vaccine-induced was based on self-reported prior COVID-19 vaccination, assuming SARS-CoV-2 sero-conversion within 5 months after receiving at least one COVID-19 vaccine was due to the vaccine.

**Table 3. Demographic, socio-economic and behavioural characteristics associated with SARS-CoV-2 infection (including asymptomatic) among sero-survey participants with ≥ 2 samples tested during December 2020-July 2021, using a Cox regression model adjusting for age and sex.**

| Characteristic | | Sero-survey participants | | SARS-CoV-2 infected | | SARS-CoV-2 age-/sex-adjusted hazard ratio | |
|---|---|---|---|---|---|---|---|
| | | N | % | N | % | HR | 95%CI |
| Age | 0–9 years | 223 | 15.9 | 35 | 12.2 | ref | |
| | 10–19 | 305 | 21.7 | 58 | 20.2 | 1.11 | 0.73–1.69 |
| | 20–29 | 161 | 11.5 | 28 | 9.8 | 0.93 | 0.56–1.53 |
| | 30–39 | 114 | 8.1 | 14 | 4.9 | 0.82 | 0.44–1.53 |
| | 40–49 | 68 | 4.8 | 15 | 5.2 | 1.11 | 0.61–2.04 |
| | 50–59 | 240 | 17.1 | 54 | 18.8 | 1.32 | 0.86–2.03 |
| | 60–69 | 202 | 14.4 | 57 | 19.9 | **1.57** | **1.03–2.39** |
| | ≥70 | 93 | 6.6 | 26 | 9.1 | 1.20 | 0.72–2.00 |
| Sex | Male | 630 | 44.8 | 128 | 44.6 | ref | |
| | Female | 776 | 55.2 | 159 | 55.4 | 0.94 | 0.74–1.18 |
| Socio-economic quintile*§ | 1st (lowest) | 116 | 14.0 | 31 | 14.5 | 1.06 | 0.68–1.65 |
| | 2nd | 152 | 18.3 | 46 | 21.5 | 1.41 | 0.95–2.09 |
| | 3rd | 161 | 19.4 | 48 | 22.4 | 1.22 | 0.82–1.81 |
| | 4th | 173 | 20.9 | 34 | 15.9 | 0.77 | 0.50–1.18 |
| | 5th (highest) | 227 | 27.4 | 55 | 25.7 | ref | |
| Education*§ | None | 502 | 54.3 | 149 | 61.3 | 1.01 | 0.57–1.79 |
| | Primary | 334 | 36.1 | 72 | 29.6 | 0.89 | 0.51–1.55 |
| | Secondary | 65 | 7.0 | 16 | 6.6 | ref | |
| | Higher | 23 | 2.5 | 6 | 2.5 | 1.13 | 0.44–2.92 |
| Health worker§ | | 27 | 4.1 | 3 | 1.9 | 0.64 | 0.28–1.45 |
| Reported comorbidities | HIV | 118 | 12 | 20 | 10.9 | 0.82 | 0.51–1.32 |
| | (history of) tuberculosis | 47 | 4.8 | 7 | 3.8 | 0.87 | 0.40–1.86 |
| | hypertension | 209 | 21.2 | 49 | 26.8 | 0.91 | 0.61–1.36 |
| | diabetes | 27 | 2.7 | 11 | 6 | 1.65 | 0.88–3.11 |
| | asthma | 59 | 6.0 | 14 | 7.7 | 1.23 | 0.71–2.12 |
| | chronic lung disease | 6 | 0.6 | 2 | 1.1 | 2.48 | 0.61–10.09 |
| | chronic heart disease | 12 | 1.2 | 1 | 0.5 | 0.40 | 0.06–2.87 |
| | leukaemia | 1 | 0.1 | 1 | 0.5 | 1.27 | 0.17–9.24 |
| BMI# | underweight | 258 | 29.5 | 52 | 26.8 | 0.72 | 0.35–1.46 |
| | normal | 352 | 40.3 | 69 | 35.6 | ref | |
| | overweight | 126 | 14.4 | 30 | 15.5 | 1.08 | 0.69–1.70 |
| | obesity | 138 | 15.8 | 43 | 22.2 | 1.25 | 0.81–1.93 |
| Smoking | non smoker | 927 | 94 | 172 | 94 | ref | |
| | (ex-)smoker | 59 | 6 | 11 | 6 | 0.67 | 0.35–1.27 |
| Public transportation | none | 587 | 59.6 | 109 | 59.9 | ref | |
| | bus/train | 395 | 40.1 | 71 | 39.0 | 0.85 | 0.63–1.16 |
| | moto taxi/shared taxi | 3 | 0.3 | 2 | 1.1 | 1.95 | 0.47–8.07 |
| Sharing bedroom | 1–2 pers. | 805 | 58.7 | 178 | 63.6 | ref | |
| | 3 or more pers. | 566 | 41.3 | 102 | 36.4 | 0.84 | 0.66–1.08 |
| Sharing toilet with other household | | 214 | 15.6 | 37 | 13.2 | 0.80 | 0.57–1.13 |
| Handwash facility | sink/faucet | 511 | 37.3 | 110 | 39.3 | ref | |
| | bucket/jar/kettle | 267 | 19.5 | 51 | 18.2 | 1.11 | 0.80–1.55 |
| | none | 593 | 43.3 | 119 | 42.5 | 0.98 | 0.75–1.27 |

(*Continued*)

**Table 3.** (Continued)

| Characteristic | | Sero-survey participants | | SARS-CoV-2 infected | | SARS-CoV-2 age-/sex-adjusted hazard ratio | |
|---|---|---|---|---|---|---|---|
| | | N | % | N | % | HR | 95%CI |
| Water available in house | yes | 581 | 42.4 | 118 | 42.1 | ref | |
| | no | 790 | 57.6 | 162 | 57.9 | 1.08 | 0.85–1.37 |

*level of education (of the mother among children and adolescents) and socio-economic level have been measured as part of the HDSS round in 2019.

§ in ≥ 18 year olds.

# in ≥16 year olds.

2021 followed relaxed non-pharmaceutical interventions for the end-of-year holidays, including partial lifting of travel restrictions, in turn resulting in many migrant workers returning from South Africa to households in Maputo. Increased (cross-border) mobility and increased social contacts coincided with the presumed introduction of the Alpha variant with increased transmissibility [13].

SARS-CoV-2 seroprevalence rose to 34.7% in June 2021, still far below the pooled 76% infection-induced seroprevalence reported in other African countries. Only a fraction of that difference in seroprevalence can be explained by different serological test specificity. Our analysis of SARS-CoV-2 sero-positivity comparing several SARS-CoV-2 antigen targets demonstrated a difference of up to 12%, thus cannot account for the twice higher seroprevalence in other sero-surveys in sub-Saharan Africa. A diagnostic performance study of the serological immunofluorescence assay supported the use of RBD and NP for IgG detection [12]. A study using the same assay however showed decreasing seroprevalence as a result of waning NP-specific IgG after three months [15]. This could explain the decreasing seroprevalence observed in July 2021.

The observed effect of age, obesity, and chronic conditions on the risk of mild disease was similar to the effect on risk of severe disease or death reported elsewhere [3,16]. While HIV and (history of) tuberculosis have been associated with COVID-19-related death [8], we observed no increased risk of infection nor of disease. Also, deprivation, increasing the risk of infection, disease and severity in several settings [3,4,17], did not affect the risk of SARS-CoV-2 infection in this population of Maputo city. In contrast, several indicators of deprivation, such as belonging to the lowest wealth quintile, lack of formal education, absence of handwash facilities, were associated with a lower risk of COVID-19. We are however hesitant to overinterpret these paradoxical findings, considering deprivation could influence voluntary self-reporting of illness, introducing a reporting bias. Among cases of respiratory illness, only anosmia, ageusia, loss of appetite and chills increased the probability of COVID-19, yet none of those symptoms was reported by more than a third of cases and the association disappeared in cases from December 2021 onwards, presumably Omicron cases. Symptoms' poor predictive value, in combination with continued reporting of other respiratory illness, hampered diagnosis of mild/moderate COVID-19 on clinical grounds only [18].

Limitations of the study include the consecutive enrolment of households which already participated in the Polana Caniço HDSS, possibly introducing a selection bias of households where an adult member is present at home, or more willing to participate in and adhere to public health measures. Second, the population within the Polana Caniço HDSS might not fully reflect the wider population of Maputo or of Mozambique, and be more homogenous. We have been careful generalising findings to those larger populations. Differences in SARS-CoV-2 risk within the study population could be smaller than those in the general population.

Third, because participating household members might tend to alter their behaviour in response to being followed-up, the so-called Hawthorne effect, we have been careful interpreting reported behaviour, and decided against describing a potentially biased number of reported contacts or reported mask wearing. Fourth, self-reporting might result in an underestimated prevalence of health conditions, e.g., HIV status, smoking. We therefore analysed health conditions only when comparing cases to the study population, in which underreporting might affect both groups similarly. Nevertheless, the association could be underestimated. Finally but importantly, self-reporting of respiratory symptoms likely suffers from underreporting, for a number of reasons including poor recall, absent household members at the time of a visit, or fear of medical procedures or healthcare costs following the reporting of symptoms. Different subgroups could also have a different likelihood to self-report symptoms. The incidence of respiratory illness and of COVID-19 should be carefully interpreted and might underestimate the actual incidence.

Active surveillance in an urban population cohort confirmed frequent COVID-19 underreporting, yet indicated that even in the first years of the pandemic, the large majority of cases were mild and non-febrile. In contrast to industrialised countries, socio-economic deprivation did not increase the risk of infection nor disease.

## Supporting information

**S1 Checklist. Inclusivity in global research.**
(DOCX)

**S1 Table.**
(DOCX)

## Acknowledgments

We thank participants of the Polana Caniço HDSS, interviewers, teams involved in laboratory testing of samples (Claudia Machume, Gercio Cuamba, Caro Van Geel), data management (Alberto Machaze, Eben Matavele, Harry van Loen), and study monitors (Dimpall Asmucrai, Carolien Hoof).

## Author Contributions

**Conceptualization:** Brecht Ingelbeen, Joachim Mariën, Anja de Weggheleire, Marianne A. B. van der Sande, Martine Peeters, Marc-Alain Widdowson, Nalia Ismael, Ivalda Macicame.

**Data curation:** Brecht Ingelbeen, Victória Cumbane, Ferão Mandlate, Barbara Barbé, Sheila Mercedes Nhachungue, Nilzio Cavele, Cremildo Manhica, Catildo Cubai.

**Formal analysis:** Brecht Ingelbeen, Victória Cumbane, Barbara Barbé, Nilzio Cavele, Audrey Lacroix, Nalia Ismael.

**Funding acquisition:** Brecht Ingelbeen, Joachim Mariën, Anja de Weggheleire, Martine Peeters, Marc-Alain Widdowson.

**Investigation:** Ferão Mandlate, Barbara Barbé, Sheila Mercedes Nhachungue, Nilzio Cavele, Cremildo Manhica, Audrey Lacroix, Philippe Selhorst, Ivalda Macicame.

**Methodology:** Brecht Ingelbeen, Victória Cumbane, Ferão Mandlate, Barbara Barbé, Nilzio Cavele, Anja de Weggheleire, Marianne A. B. van der Sande, Martine Peeters, Nalia Ismael, Ivalda Macicame.

**Project administration:** Ferão Mandlate, Barbara Barbé, Cremildo Manhica, Ivalda Macicame.

**Supervision:** Brecht Ingelbeen, Victória Cumbane, Ferão Mandlate, Barbara Barbé, Nilzio Cavele, Cremildo Manhica, Neusa Maimuna Carlos Nguenha, Audrey Lacroix, Joachim Mariën, Marc-Alain Widdowson, Nalia Ismael, Ivalda Macicame.

**Validation:** Victória Cumbane, Ferão Mandlate, Barbara Barbé, Nilzio Cavele, Cremildo Manhica, Catildo Cubai, Neusa Maimuna Carlos Nguenha, Audrey Lacroix, Philippe Selhorst, Nalia Ismael.

**Visualization:** Brecht Ingelbeen.

**Writing – original draft:** Brecht Ingelbeen, Victória Cumbane.

**Writing – review & editing:** Brecht Ingelbeen, Victória Cumbane, Ferão Mandlate, Barbara Barbé, Sheila Mercedes Nhachungue, Nilzio Cavele, Cremildo Manhica, Catildo Cubai, Neusa Maimuna Carlos Nguenha, Audrey Lacroix, Joachim Mariën, Anja de Weggheleire, Esther van Kleef, Philippe Selhorst, Marianne A. B. van der Sande, Martine Peeters, Marc-Alain Widdowson, Nalia Ismael, Ivalda Macicame.

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
