## [Decision Letter · Decision Letter 0]

10 Jun 2024

PGPH-D-24-00678

Mild and moderate COVID-19 during Alpha, Delta and Omikron pandemic waves in urban Maputo, Mozambique, December 2020-March 2022: a population-based surveillance study

Dear Dr. Ingelbeen,

Thank you for submitting your manuscript to PLOS Global Public Health. After careful consideration, we feel that it has merit but does not fully meet PLOS Global Public Health’s publication criteria as it currently stands. Therefore, we invite you to submit a revised version of the manuscript that addresses the points raised during the review process.

Please ensure the 3 necessary areas (sampling, epi comparison and study limitations) is adequately addressed, in the revised manuscript. The other recommended corrections raised by the reviewers should be carefully assessed and provide adequate explanation in the rebuttal should revision not possible or warranted.

We look forward to receiving your revised manuscript.

Kind regards,

Nei-yuan (Marvin) Hsiao

Academic Editor

Journal Requirements:

2. Please include a complete copy of PLOS’ questionnaire on inclusivity in global research in your revised manuscript. Our policy for research in this area aims to improve transparency in the reporting of research performed outside of researchers’ own country or community. The policy applies to researchers who have travelled to a different country to conduct research, research with Indigenous populations or their lands, and research on cultural artefacts. The questionnaire can also be requested at the journal’s discretion for any other submissions, even if these conditions are not met. Please find more information on the policy and a link to download a blank copy of the questionnaire here: https://journals.plos.org/globalpublichealth/s/best-practices-in-research-reporting. Please upload a completed version of your questionnaire as Supporting Information when you resubmit your manuscript.

Additional Editor Comments (if provided):

The study provides valuable insight into SARS-CoV-2 in Mozambique where there is a relatively paucity of COVID-19 epidemiological data. The reviewers had provides valuable comment which though minor, can significantly strengthen the manuscript.

Please pay particular attention to the following area when addressing the reviewer's comments:

1. Sampling strategy and its impact on interpretation of finding.

2. In discussion, the difference of the wave/epi pattern of Mozambique with its neighbouring countries warrant some comments.

3. A robust study limitations section, particularly related to sampling strategy, is needed.

Reviewers' comments:

Reviewer's Responses to Questions

**Comments to the Author**

1. Does this manuscript meet PLOS Global Public Health’s publication criteria? Is the manuscript technically sound, and do the data support the conclusions? The manuscript must describe methodologically and ethically rigorous research with conclusions that are appropriately drawn based on the data presented.

Reviewer #1: Yes

Reviewer #2: Yes

2. Has the statistical analysis been performed appropriately and rigorously?

Reviewer #1: Yes

Reviewer #2: Yes

3. Have the authors made all data underlying the findings in their manuscript fully available (please refer to the Data Availability Statement at the start of the manuscript PDF file)?

Reviewer #1: Yes

Reviewer #2: Yes

4. Is the manuscript presented in an intelligible fashion and written in standard English?

Reviewer #1: Yes

Reviewer #2: Yes

5. Review Comments to the Author

Reviewer #1: Review of

Mild and moderate COVID-19 during Alpha, Delta and Omikron pandemic waves in urban Maputo, Mozambique, December 2020-March 2022: a population-based surveillance study

General comments:

This paper presents a epidemiological study about mild and moderate COVID-19 occurrence in Mozambique proceeded through a population-based surveillance

General comments

Thank you for stating the following financial disclosure and Acknowledgments Section and to provide a repository information for your data, codes and maps.

Please, check if the keywords were included in DeCS/MeSH

Introduction

Thanks for the introduction, it is very well written, easy to read and presents what is necessary on the topic. Authors may choose to discreetly include information about study locations, for example, population size.

Please check grammar of sentence in third paragraph. Use commas to numbers.

I understood that in Sep 2021 5% received at least 1 dose, and that 6 months later there was an increase to 40% with 1 dose, is this correct? It would be good to place more emphasis on vaccination coverage.

Why Omicron was spelled with K, not C (for English)?

Methods

In the second paragraph it would be delimited as a section "data collection" better describing the tools for the two studies (i) socio-economic questionnaire, (ii) questionnaire to collect symptoms and (iii) self-administered biological samples. Please also explain whether the questionnaires were different for each age group.

Were questions about vaccination collected or did population coverage be used to interpret the results?

I suggest that the authors describe better the definition of suspected cases (possible cases) including a sentence answering this question: What happens if the covid case is not confirmed, is it a negative PCR that was used to confirm the case or rule it out? Even with a well-reported clinic? see an example of how well you defined seropositivity (line 110). Even though the data analysis section perfectly indicates procedures to do that, it would be good to define it in a simpler way here.

It was not clear to me whether, when there was a positive case, other residents of the same residence were tested even if they were asymptomatic, close contact was not a possible case? Can you explain this better here, in the methods section before appears in the next sections?

Can you explain why close contact was not a possible case? I understand that this is mild and moderate covid, but it would be nice to inform you if there were asymptomatic cases or if this was not considered in this study. As you pointed in the discussion.

Laboratory procedures

Laboratory procedures is a little difficult to follow because it is a very long paragraph, I suggest dividing and removing unnecessary punctuation that divides sentences like ( : ) on line 100.

I missed the reference of the collections in each study, is difficult to follow, especially in time.

What was done with the samples after they were tested? Stored? for how long, destroyed?

Line 99, move citation to reference, following the journal's rules. Name (year) [ref]

Data analysis

It's very good! It's very good! I only have a question, is the prevalence adjusted for the tests sensitivity/specificity?

Previous, the methods need specifying better the outcomes. Define and re-specify your outcomes, presenting all your variables in clear terms.

Results

You can include a comma in the numbers presented in the text, it is easier to read. (line 149-153)

HIV diagnosis or comorbidities were self-reported? Please include this information in the methods section as a case definition.

In line 151-152, I don't understand who recorded socio-economic status, does information only apply to over 18yo for example? Please move this information, clarifying this doubt, to the methods section. Please answer the following question very simply: Considering that the study includes people of all ages, are the educational level results for those over 18? (line152-153)

Figure 1, can you explain why some people were not tested?

Please, include in all foot tables the name of the test used in the presented aOR, OR, HR (...)

Table 1 - replace throat pain for sore throat.

Figure 2, It is not very clear if “positive after vaccination” refers to vaccine-induced antibody +, please, consider replace the text for meaning is clear.

Line 181 there is no need to parentheses.



Discussion/Conclusion

Thanks for the discussion section, its drawn appropriately based on the data presented.

Please remove the parentheses of (symptomatic) on line 219, the sentence is confusing.

Review line 253

Did the authors observe any limitation? It would be nice to comment on in the discussion section. Please explain whether these limitations have been mitigated and how you plan to address them. Otherwise, authors can discuss any participation issue in the limitations section.

There are few spelling errors that need to be revised.

Reviewer #2: While interest in SARS-CoV-2 has waned with time, the manuscript provides valuable insight into trying to understand the impact of the pandemic in sub-Saharan Africa, and warrants publication. There are however, a few issues that need to be addressed:

1. It is unclear how participants were selected from the HDSS for the 2 weekly follow-ups, and what if any sampling strategy was used to recruit them.

2. An introduction to the HDSS population and description of the setting would be useful. My concern is that if the population were very homogeneous in terms of socioeconomic status, it would be difficult to pick up any differences in SARS-CoV-2 epidemiology with the current population size.

3. The aHR values for socioeconomic quintiles and education in Table 3 are missing. These results are crucial to the findings of the study, and need to be reported.

4. Neighboring South Africa experienced a massive wave of SARS-Cov-2 infections in June/July 2020, and again in December 2020. As official case numbers under-counted the real extent of the pandemic, is it possible that Mozambique experienced mild SARS-CoV-2 in the middle of 2020, before the survey was started, and with the waning of antibody levels this "first" wave was missed? Either way, further interrogation of the lower seroprevalence results found in Maputo is needed, as the authors do not provide a sufficient explanation. The argument that prolonged NPI prevented transmission (line 233), might in true with regards to other neighbouring countries, but in the reference no 14 South Africa is noted to have stricter policies than Mozambique, contradicting the authors' explanation.

5. Line 252-253: "This discrepancy would be in line with observed lower pathogenicity in other sSA studies" - reference is needed.

6. Line 247-248: No relationship between HIV and COVID-19 death was found. But how was HIV status ascertained? Was it only through respondent self-reporting, in which case exposure status misclassification might underestimate the relationship. I am not sure what the current HIV prevalence for adults in Mozambique is, but presume it is higher than the 7.5% reported amongst study participants.

6. PLOS authors have the option to publish the peer review history of their article (what does this mean?). If published, this will include your full peer review and any attached files.

**Do you want your identity to be public for this peer review?** For information about this choice, including consent withdrawal, please see our Privacy Policy.

Reviewer #1: **Yes: **Fabiana Ganem

Reviewer #2: **Yes: **Hannah Hussey

While revising your submission, please upload your figure f

---

## [Editor Report · Decision Letter 1]

10 Jul 2024

Mild and moderate COVID-19 during Alpha, Delta and Omicron pandemic waves in urban Maputo, Mozambique, December 2020-March 2022: a population-based surveillance study

PGPH-D-24-00678R1

Dear Dr Ingelbeen,

We are pleased to inform you that your manuscript 'Mild and moderate COVID-19 during Alpha, Delta and Omicron pandemic waves in urban Maputo, Mozambique, December 2020-March 2022: a population-based surveillance study' has been provisionally accepted for publication in PLOS Global Public Health.

Best regards,

Nei-yuan (Marvin) Hsiao

Academic Editor